# DDX20: A Multifunctional Complex Protein

**DOI:** 10.3390/molecules28207198

**Published:** 2023-10-20

**Authors:** Lu He, Jinke Yang, Yu Hao, Xing Yang, Xijuan Shi, Dajun Zhang, Dengshuai Zhao, Wenqian Yan, Xintian Bie, Lingling Chen, Guohui Chen, Siyue Zhao, Xiangtao Liu, Haixue Zheng, Keshan Zhang

**Affiliations:** 1State Key Laboratory for Animal Disease Control and Prevention, College of Veterinary Medicine, Lanzhou University, Lanzhou Veterinary Research Institute, Chinese Academy of Agricultural Sciences, Lanzhou 730000, China; 2Gansu Province Research Center for Basic Disciplines of Pathogen Biology, Lanzhou 730046, China

**Keywords:** DDX20, NF-κB, transcription, SMN complexe, cancer, miRNA

## Abstract

DEAD-box decapping enzyme 20 (DDX20) is a putative RNA-decapping enzyme that can be identified by the conserved motif Asp–Glu–Ala–Asp (DEAD). Cellular processes involve numerous RNA secondary structure alterations, including translation initiation, nuclear and mitochondrial splicing, and assembly of ribosomes and spliceosomes. DDX20 reportedly plays an important role in cellular transcription and post-transcriptional modifications. On the one hand, DDX20 can interact with various transcription factors and repress the transcriptional process. On the other hand, DDX20 forms the survival motor neuron complex and participates in the assembly of snRNP, ultimately affecting the RNA splicing process. Finally, DDX20 can potentially rely on its RNA-unwinding enzyme function to participate in microRNA (miRNA) maturation and act as a component of the RNA-induced silencing complex. In addition, although DDX20 is not a key component in the innate immune system signaling pathway, it can affect the nuclear factor kappa B (NF-κB) and p53 signaling pathways. In particular, DDX20 plays different roles in tumorigenesis development through the NF-κB signaling pathway. This process is regulated by various factors such as miRNA. DDX20 can influence processes such as viral replication in cells by interacting with two proteins in Epstein–Barr virus and can regulate the replication process of several viruses through the innate immune system, indicating that DDX20 plays an important role in the innate immune system. Herein, we review the effects of DDX20 on the innate immune system and its role in transcriptional and post-transcriptional modification processes, based on which we provide an outlook on the future of DDX20 research in innate immunity and viral infections.

## 1. Introduction

DDX20 is also known as Gemin3 or DP103 and is among the most extensively researched genes [1]. As a member of the DEAD-box family, DDX20 plays a crucial role in RNA metabolism by acting as an RNA-unwinding enzyme [2]. Other studies have reported that DDX20 plays a crucial role in influencing innate immunity by utilizing miRNAs to regulate NF-κB signaling pathway. This, in turn, can result in the development of tumors and inflammatory diseases. Previous studies have reported that DDX20 impacts early embryonic development and plays a role in regulating ovarian morphology and function [3]. A study has shown that DDX20 is crucial for the differentiation of oligodendrocytes and maintenance of myelin gene expression [4]. Additionally, DDX20 serves as an Olig2-binding protein, which helps in maintaining the function of neurons, oligodendrocytes, and histone cells in the nervous system. Therefore, DDX20 significantly affects the central nervous system [4,5]. Furthermore, increasing evidence suggests that DDX20 plays a crucial role in predicting the development, invasion, and drug response of cancer. In antiviral natural immunity, DDX20 inhibits the replication of vesicular stomatitis virus (VSV) and herpes simplex virus type 1 (HSV-1) by inducing the production of interferon beta type I (IFN-β) [2]. This review will systematically describe the research progress of DDX20 from various aspects and detail the research progress of DDX20 in terms of neurological- and tumor-related aspects to provide a reference for antiviral and tumor research.

## 2. Distribution, Structure, and Subcellular Localization of DDX20

As a novel human member of the DEAD-box family with ATP-dependent RNA-unwinding enzymes, DDX20 was detectable at the molecular level in mammalian cells with monoclonal antibodies against DDX20 elevated to 103 kDa in size [6]. Subsequent studies revealed that the Cryptobacterium hidradenum gene Mel-46 encoding the DDX20 protein is expressed throughout its development and plays a facilitating role during development [7]. The monitoring results at the cellular level revealed that decreased DDX20 expression could be detected in cancerous tissues of hepatocytes, thus affecting the disease process [8,9]. In addition, at the tissue and organ level, DDX20 was expressed in the testis and even in steroid- and nonsteroid-producing tissues [10]. Furthermore, DDX20 is significantly overexpressed in certain cancerous tissues, such as hepatocellular carcinoma and colorectal, prostate, and gastric cancers, and usually indicates a good prognosis [11,12,13,14]. DDX20 was first encoded by the genome of the parthenogenetic multicellular organism Dictyostelium discoideum and is retained in postnatal animals [15]. Some drugs, such as statins, can also inhibit DDX20 expression through the mevalonate pathway and downstream of RhoA [16].

The DEAD-box RNA-decapping enzyme family features a unique Asp–Glu–Ala–Asp (D-E-A-D) sequence motif [17]. All DEAD-box decapping enzymes, including DDX20, possess a conserved core structural domain at the N-terminal end, primarily comprising two recombinase A (RecA)-like structural domains [18]. As shown in Figure 1, this core structural domain includes nine conserved motifs, namely Q, I, Ia, Ib, II, III, IV, V, and VI, which are involved in ATP binding and hydrolysis, RNA substrate binding, regulating decapping enzyme activity via ATP-binding hydrolysis, and regulating ATPase activity [1]. The D-E-A-D sequence motif is primarily found in the highly conserved motif II, which is also the source of the “DEAD-box” name. Nonconserved N- and C-terminal auxiliary domains flank the core domain, ranging in size from a few to several hundred amino acids. These are mainly associated with specific functions of the decapping enzyme [19]. For instance, the C-terminal region of DDX20 is necessary for it to maintain its uncoupling helicase activity [20].

DDX20 participates in a complex biological process regarding its subcellular localization. As a member of the SMN complex, DDX20 plays a role in the assembly of snRNP [21]. The assembly process of snRNP involves a transition in localization from the cytoplasm to the nucleus, and DDX20, correspondingly, undergoes a change in position during this process [22]. Before engaging in the snRNP assembly process, DDX20 initially coassembles with other proteins in the cytoplasm to form the complete SMN complex. The SMN complex is widely present in the cytoplasm, excluding muscle cells. Subsequently, DDX20 coassembles SmB, SmD2, and SmD3 to snRNA-specific sites in the cytoplasm [23].

Upon assembly and capping at the 5′ end, snRNP enters the nucleus aided by the SMN complex and the import factor importin β, with DDX20 associated with snRNP following suit [24,25]. In the nucleus, the SMN complex cotargets a subnuclear organelle known as Cajal bodies (CBs) with snRNP but separates from snRNP shortly thereafter [26,27].

The localization of DDX20 in the cytoplasm and nucleus is subject to several influencing factors. First, the SMN complex undergoes extensive phosphorylation modifications, and dephosphorylation modifications of several proteins within the SMN complex, including DDX20, obstructs the movement of the SMN complex into the nucleus and its accumulation in the CBs [28]. Second, the transition of the SMN complex from the cytoplasm to the nucleus under normal conditions necessitates intact sumoylation modification. The impaired coupling of SUMO, similar to phosphorylation modification, can curtail the nuclear localization of the SMN complex, leading to its abnormal cytoplasmic accumulation [29]. Interestingly, SMN proteins can serve as specific targets for the acetyltransferase CBP, and an acetylation site, K119, has been identified. However, unlike the previous two modifications, acetylation promotes the movement of SMN to the nucleus and augments its cytoplasmic localization [30]. Lastly, Gemin4, a member of the SMN complex, contains three putative nuclear localization signal (NLS) motifs that propel the movement of other Gemin proteins in the SMN complex from the cytoplasm into the nucleus, disrupting the subnuclear localization of the Cajal-body–tagged protein coilin in a dose-dependent manner [31]. Therefore, DDX20, which is heavily localized in the cytoplasm, enters the nucleus with the SMN complex as it functions, a process that can be swayed by an array of factors.

## 3. Splicing Features of DDX20

Existing research indicates that the motor neuron protein SMN, along with Gemin2–8 proteins, including DDX20/Gemin3 and Unrip, form a stable SMN complex, with DDX20/Gemin3 playing a vital role in the assembly of this complex [32,33,34]. The assembly process of the SMN complex is modular, involving the physical association of several SMN complex proteins. SMN/Gemin8/Gemin7, situated at the core of the complex, recruit other proteins to form the complete assembly [34].

As a molecular companion, the SMN complex facilitates the assembly and function of spliceosomal small ribonucleoprotein (snRNP) [35]. snRNP comprises U-rich small nucleus RNA (snRNA) and 7 small (Sm) or Sm-like (LSm) proteins assembled into a complete stem ring structure in the cytoplasm [34,36]. This assembly process, which involves multiple SMN complex components, necessitates a cellular positional shift.

Initially, snRNAs (excluding U6) are transcribed in the nucleus by RNA polymerase II as precursor RNAs, containing an additional stem-loop structure at the 3′ end and a monomethylated m7GpppG (m7G) cap structure at the 5′ end [37]. This precursor snRNA subsequently binds to an snRNA export complex comprising multiple proteins through its 5′ cap structure, leading to its export into the cytoplasm [38].

In the cytoplasm, Sm and Sm-like proteins first bind to the chloride conductance regulatory protein (pICln) and protein arginine methyltransferase 5 (PRMT5) complexes. These proteins are then prearranged into spatial positions and separated upon the addition of the SMN complex, which assembles the Sm and Sm-like proteins into protein binding sites on snRNAs, forming the complete snRNP with a seven-membered loop structure [39,40].

Finally, the snRNA with the Sm core undergoes methylation at its 5′ end, catalyzed by trimethylguanosine synthase 1 (TGS1), to form an m3G cap structure. This process is mediated by the SMN complex and importin β, facilitating the performance of splicing functions in the nucleus [15,41]. This assembly process is complex and involves different components playing unique roles. The focus of this section is a brief overview of the role of DDX20/Gemin3 in this process.

First and foremost, the SMN complex serves a critical, nonreplaceable role in snRNP assembly, and DDX20/Gemin3 is indispensable for the stabilization of the complex. Within the SMN complex, DDX20/Gemin3 engages with SMN proteins, along with Gemin2, 4, and 5. This interaction recruits Gemin3 as a core component of the SMN complex, promoting its stability [42,43,44]. A sumoylation modification of the core components of the SMN complex, including Gemin3, is necessary for this interaction. A desumoylation modification of Gemin3 could decrease binding to several core proteins and potentially impact the cellular localization SMN complex [29]. As shown in Figure 2, Gemin3 forms an SMN complex with several other proteins and there is interaction between the various proteins.

The constituent proteins of the SMN complex mutually interact; thus, changes in one protein affect the stability and abundance of other proteins. For instance, a reduction in SMN protein results in the disappearance of Gems and a decrease in several Gemin levels, ultimately affecting snRNP assembly [45] and thereby indicating the potential role of DDX20/Gemin3 in maintaining the stability of the SMN complex.

Second, the SMN complex is extensively regulated by phosphorylation. DDX20/Gemin3, containing 13 phosphorylation sites, is among the most highly phosphorylated proteins in the complex. The phosphorylation of DDX20/Gemin3 and other complex proteins influences the exchange of the SMN complex between the cytoplasm and nucleus as well as the assembly of snRNP [28]. The effects of DDX20/Gemin3 in the SMN complex extend to biological systems. Its deletion in the Drosophila mesoderm and muscle layer can lead to impaired locomotion, severe developmental defects, and even larval death and is potentially associated with SMA caused by SMN protein mutations [46].

Existing studies have reported that Gemin3 significantly impacts snRNP assembly. Notably, the disruption of snRNP assembly following Gemin3 knockdown via RNAi provides the most direct evidence [47,48]. The vSMN complex can bind to Sm proteins and facilitate their assembly into complete snRNPs by binding to snRNAs. The fact that Gemin3 can interact with SmB, SmD2, and SmD3 in Sm proteins suggests its role in snRNP biogenesis [23].

Further evidence regarding the role of Gemin3 in snRNP assembly is provided by Almstead et al., who found that Gemin3′s specific cleavage by the poliovirus-encoded protease 2Apro results in reduced Gemin3 protein levels, which led to specific rearrangement of Sm proteins from the nucleus to the cytoplasm and a reduction in snRNP assembly [47]. In addition, Gemin3 genetically and physically interacts with two factors in snRNP assembly, namely pI-Cln and Tgs1. Deleting pICln and Tgs1 would result in the same viability and motility phenotype as a Gemin3 deficiency [49]. The phenotypic defect would be exacerbated by the disruption of two amyotrophic lateral sclerosis (ALS) genes encoding the proteins TDP-43 and FUS. Such results offer additional evidence regarding the role of DDX20 in snRNP assembly and motor neuron diseases [50]. Figure 3 illustrates the assembly of the SMN complex as well as the process of translocation.

Based on the outlined role of DDX20/Gemin3, we understand that it contributes to RNA splicing not through direct interaction with the pre-RNA but by playing a critical part in the assembly of the spliceosomal snRNP as a component of the SMN complex. This process facilitates the subsequent modification and splicing of the pre-RNA by the snRNP within the nucleus. In this intricate and indirect manner, DDX20/Gemin3 is instrumental in the RNA splicing process.

## 4. DDX20 Represses Transcription by Inhibiting Transcription Factors

In recent years, the role of DDX20 in transcription has been increasingly reported and considerable evidence regarding its role as a transcription regulator has been reported. DDX20 primarily suppresses transcription by interacting with the nuclear receptor steroidogenic factor-1 (SF-1) [51]. SF-1, a member of the transcription factor nuclear receptor superfamily, is a crucial regulator of the hypothalamic–pituitary–gonadal axis and the endocrine factor of the adrenal cortex [52]. DDX20 is highly expressed in steroid-producing tissues, which also highly express SF-1 [10]. DDX20 directly interacts with the C-terminal inhibitory domains of SF-1 via its nonconserved C-terminal domains to inhibit the transcriptional activity of SF-1 [20]. Further studies have elaborated on the specific mechanisms by which DDX20 interacts with SF-1 and inhibits its transcriptional activity. The transcriptional activity of SF-1, a transcription factor, is affected by other transcription factors, coregulatory factors, and post-translational modifications [53]. Sumoylation, a ubiquitin-like modification that occurs following SF-1 translation, inhibits the ability of SF-1 to activate target gene expression [54]. DDX20 enhances the sumoylation of SF-1 mediated by protein inhibitor of activated STATs proteins (a type of E3-SUMO ligase) after direct interaction with the SF-1 protein to inhibit its transcriptional activity [55]. This interaction also facilitates the relocalization of SF-1 to discrete nucleosomes or lesions. Exploring this inhibitory mechanism revealed that Histone deacetylase is involved in sumoylation modification of transcription factor SF-1 and interacts less with the corepressor; however, it may function as an E3 ligase during sumoylation [56].

SF-1 has a homologous gene in arthropods, known as FTZ-F1 (fushi tarazu factor-1) [57]. A study confirmed that DDX20 interacts with FTZ-F1 in paralfalfa via a yeast two-hybrid assay and reported that DDX20 can suppress FTZ-F1 expression via a mechanism similar to that of SF-1 inhibition by RNAi [58]. FTZ-F1 is closely related to vitellogenin (VTG) and is involved in the development of vitelline in the ovary and can influence the secretion of several endocrine hormones [59]. The inhibition of FTZ-F1 via DDX20 eventually affects ovarian development.

The Forkhead transcription factor (FOXL) 2, a transcription factor closely related to FTZ-F1 and DDX20, also plays a significant role in regulating ovarian development. In the ovary, FOXL2 is involved in regulating cholesterol and steroid metabolism, apoptosis, reactive oxygen species detoxification, and cell proliferation [60]. Initially, it was discovered that DDX20 interacted with FOXL2 and their coexpression in cells enhanced FOXL2-mediated ovarian cell death [61]. Subsequent studies confirmed that FOXL2 interacts with DDX20 and FTZ-F1 and not only downregulates VTG synthesis by regulating follicular cell apoptosis through DDX20 but also potentially regulates the steroidogenic pathway through FTZ-F1.

In addition to F-1, DDX20 has been discovered to interact with two other transcription factors to inhibit their activity by different mechanisms compared with that used with SF-1. First, DDX20 can engage with the mitotic Ets transcription inhibitor (PE-1/METS) via its C-terminal domain. Ets is a transcription factor that serves as a nuclear target to activate the Ras–MAPK signaling pathway, while PE-1/METS, another member of the Ets family, acts as an Ets inhibitor to repress the Ras-dependent proliferation of Ets target genes, thus causing macrophage growth arrest [62].

DDX20 interacts with the N-terminal domain of PE-1/METS via its C-terminal domain, which has an inhibitory effect. In this process, DDX20 also recruits factors such as histone deacetylase HDAC-2 and HDAC5 [63]. However, interestingly, this transcriptional inhibition mediated by DDX20 targets only individual promoters, such as c-myc and cdc2, without affecting Ets ternary complex-facilitated transcription [64].

The C-segment domain of DDX20 also interacts with another transcription factor known as early growth reaction 2 (Egr2/Krox-20). All proteins interacting with the C-terminal of DDX20 are summarized in Table 1. Egr2/Krox-20 is one of four members of the early growth response gene family and is associated with peripheral nervous system myelination and posterior brain development [65].

Thus, DDX20 does not undergo sumoacylation as DDX5 and its transcription regulation activity do not solely rely on a single intrinsic function but involve multiple mechanisms, many of which depend on its unique noncore C-terminal domain. This multifaceted approach to transcriptional regulation reinforces the complexity of the function of DDX20 in this essential cellular process.

## 5. Biogenesis of DDX20 and miRNA

DDX20/Gemin3 also contributes to the maturation of miRNA. The biogenesis of miRNA begins with the transcription of miRNA genes into primary miRNA (pri-miRNA) by RNA polymerase II. The pri-miRNA is subsequently processed into a precursor miRNA (pre-miRNA) of ~70 nucleotides by a complex containing the RNase III endonuclease Drosha and a protein containing a double-stranded RNA-binding structural domain (dsRBD) named Pasha (DGCR8) [77,78].

Subsequently, the stem–loop-structured pre-miRNAs are transported into the cytoplasm, which is dependent on the GTP-dependent transport protein Exportin-5 (Exp5) [79]. In the cytoplasm, the RNase III endonuclease Dicer processes pre-miRNA into an siRNA–miRNA duplex of ~22 nucleotides. The mature miRNA strand is subsequently retained in the RNA-induced silencing complex (RISC) along with other proteins [80].

Although most Gemin3 and Gemin4 are components of the SMN complex, the complexes of Gemin3 and Gemin4 isolated from HeLa and neuronal cells are not part of the SMN complex. Instead, these complexes coprecipitate with polyribosomes [71,81,82]. Numerous studies have established the connection of Gemin3/DP103 and Gemin4 with the Argonaute (Ago) protein family member Argonaute [60,83,84]. Figure 4 shows us the mIRNAde1 maturation process and the role Gemin3 plays in it.

MiRNAs can inhibit the translation of partially complementary target messenger RNAs by directing the sequence-specific degradation of both fully and partially complementary target mRNAs [85,86]. However, regarding the specific role of Gemin3 in miRNA biogenesis, researchers suggest that Gemin3 is a member of RISC, given the presence of the Gemin3 protein in the peripheral axons of the mouse sciatic nerve and its capacity to form a multiprotein RISC in response to specific treatment [72].

RISC was not previously identified as having decapping components for siRNAs and miRNAs within the complex. Consequently, researchers posit that Gemin3 serves as a decapping enzyme within the RISC complex and plays a role in RNA decapping or recombination during miRNA maturation as well as in target RNA recognition [72].

## 6. Functions of DDX20 in the Innate Immune Signaling Pathway

### 6.1. Effect of DDX20 on the NF-κB Signaling Pathway

The NF-κB family comprises six distinct components. When activated, they produce various proinflammatory factors that regulate inflammation and significantly contribute to innate immunity [87,88]. The NF-κB signaling pathway controls the expression of proinflammatory cytokines and anti-infective factors, including TNF-α, interleukin-1 (IL-1), IL-6, IL-8, adhesion molecules, and cc chemokine ligand 5 (CCL5) [89]. In addition, NF-κB signaling pathway governs cellular processes such as cell proliferation, differentiation, and apoptosis [90,91].

While the NF-κB signaling pathway has an antiapoptotic role, its dysregulation has been implicated in the pathogenesis of most human malignancies [92]. Therefore, manipulating the NF-κB signaling pathway can provide a path for developing novel methods to combat diseases such as cancer [93]. For instance, alterations in DDX20 expression in cancer tissues can affect NF-κB activity, leading to cancer development [94].

DDX20 can have two distinct effects on the NF-κB signaling pathway. Numerous studies have reported that DDX20 does not directly affect NF-κB activity but rather acts through a naturally occurring small non-coding RNA (miRNA) intermediate. The present study demonstrated that DDX20 can modulate the signaling of the NF-κB pathway through miRNA-22, miRNA-140-3p, and miR-222 [8,95]. Additionally, miR-361-5p was found to regulate DDX20, thereby influencing NF-κB pathway signaling [13]. First, DDX20 can inhibit NF-κB activity. DDX20 and miRNAs together form a ribonucleoprotein complex, and miRNA library screening has revealed that several miRNAs can inhibit NF-κB activation. This inhibition prevents the activation of the NF-κB signaling pathway [8]. It has been well established that miRNAs can inhibit NF-κB activity by regulating the expression of two NF-κB coactivators, namely nuclear receptor coactivator protein 1 (NCOA1) and nuclear receptor-interacting protein 1 (NRIP1) [96]. Conversely, DDX20 can also enhance NF-κB activity, notably by impairing miRNA function by decreasing its own expression, leading to impaired NF-κB inhibitory miRNA function [8,97].

Another way DDX20 can inhibit the NF-κB signaling pathway is through miRNA-140 dysfunction, which increases expression of its downstream target gene Dnmt1 and the methylation of CpG islands in the promoter region of metallothionein (MTs) and decreases MT expression, leading to enhanced NF-κB activity [9]. Second, DDX20 can boost the NF-κB signaling pathway by enhancing the phosphorylation of TAK 1, where it acts as a cofactor of TAK1, thus enhancing the activity of the NF-κB signaling pathway [98]. More details regarding this aspect will be discussed in the subsequent sections.

### 6.2. DDX20 Affects p53 Signaling Pathway Conduction

The TP53 gene encodes the key p53 transcription factor and evolutionarily conserved tumor suppressor involved in maintaining genomic stability [99]. To accomplish this, it activates DNA repair responses and initiates apoptosis in damaged host cells [100]. Its activation controls core programs such as cell cycle arrest and apoptosis [101]. In addition, p53 is closely related to immune responses as well as various inflammatory diseases [102].

In fact, the effect of DDX20 on the p53 signaling pathway can be realized by directly affecting the pivotal p53 protein. Changes in DDX20 expression can consequently impact the organism’s state through the p53 signaling pathway. Reportedly, DDX20 interacts with p53 protein through its C-terminal structural domain [1]. The normal expression of DDX20 stabilizes motor neurons and uses genomic stability and regulates Mdm2 splicing to restrain the p53 signaling pathway, thereby preserving normal neural development [4].

In this context, several cytokines, such as oligodendrocyte transcription factor 2 (Olig2) and Epstein–Barr (EB) nuclear antigen 3C (EBNA3C), can directly interact with DDX20 to stabilize its expression. This interaction inhibits transcription and apoptosis caused by the p53 signaling pathway and its downstream genes within the organism [4,67].

## 7. DDX20 Plays Different Roles in Cancers through the NF-κB Signaling Pathway

DDX20 not only plays a role in the invasion of multiple pathogens but also plays an active role in the case of multiple tumorigenesis. We have previously mentioned that DDX20 is inconsistently expressed in different cancerous tissues and studies have reported that DDX20 exhibits contradictory effects in cancer. For example, it plays an oncogenic role in breast, prostate, and liver cancers while acting as a tumor suppressor [103,104,105].

In breast cancer, DDX20 is involved in cell signaling pathway activity, which is a key factor in tumorigenesis. The NF-κB signaling pathway is more closely linked to tumor development, and, reportedly, it can improve cancer cell survival, promote cancer cell angiogenesis and migration, and has other characteristics alongside being associated with the poor prognosis of cancer diseases [106]. It is because of the role of the NF-κB signaling pathway in tumor growth that it can be used to inhibit carcinogenesis [107]. DDX20 is an important cofactor for the phosphorylation of transforming growth factor-β-activated kinase-1 (TAK1) by NF-κB-activated IκB kinase 2 (IKK2), enhancing the activity of the NF-κB signaling pathway by stimulating TAK1 phosphorylation [108]. TAK1 is a member of the mitogen-activated protein kinase (MAPK) family that plays a key regulatory role as an upstream component of the NF-κB signaling pathway [109]. Enhanced NF-κB signaling pathway activation increases in the expression of two downstream products, matrix metalloproteinase 9 (MMP9) and multidrug resistance gene 1 (MDR1), and, as the chief role of MMP9 is to degrade the extracellular matrix, this change further increases tumor metastasis and drug resistance [19,110]. Conversely, enhanced NF-κB activity and increased downstream MMP9 expression would also lead to increased DDX20 expression. Thus, the establishment of the DDX20–NF-κB–MMP9 axis could better reveal the mechanism by which DDX20 can promote cancer development [69]. In addition, DDX20 may exhibit an miRNA-processing role in breast cancer. A group of studies reported that DDX20 exhibited a negative correlation with an miRNA, namely miR-222, suggesting that DDX20 affects the NF-κB activity through miR-222 to promote breast cancer development [95]. Based on this property of DDX20 in breast cancer, researchers believe that DDX20 can serve as an active alternative to certain anticancer drugs. DDX20 enhances the sumolylation modification of YAP, thereby increasing YP-TEAD dependence and statin sensitivity in patients with triple-negative breast cancer [16]. Statins, such as simvastatin (SMV), are cholesterol-reducing lipophilic statins that inhibit DDX20 expression by inhibiting 3-hydroxy-3-methylglutaryl-CoA reductase (HMGCR), which is positively associated with DDX20 [103]. Figure 5A demonstrates that DDX20 affects tumorigenesis and development through the NF-κB signaling pathway. Further studies have reported that simvastatin downregulates DDX20 not only through the classical mevaleric acid pathway and the downstream component of RHoA but also through the miRNA-mediated nonclassical pathway [111,112]. Simvastatin ultimately inhibits breast cancer by decreasing DDX20 expression. In addition to its role through the NF-κB signaling pathway and miR-222, DDX20 acts through the cellular redox pathway and Wingless/Integrated(Wnt) signaling pathway. In cancer stem cells (CSCS), DDX20 drives a positive feedback loop wherein DDX20 promotes its transcriptional regulation via transcription factor 4 (TCF 4) to stimulates the aggressiveness of breast cancer via Wnt/beta-catenin signaling [113]. Another example of the carcinogenic effect of DDX20 is in prostate cancer. High DDX20 expression in the tumor tissue of patients with prostate cancer also enhances tumor growth and metastasis via the DDX20–NF-κB–MMP9 axis [11]. These results suggest a role of DDX20 in tumor metastasis.

In contrast to its role in prostate and breast cancers, DDX20 acts as a tumor suppressor in liver cancer and suppresses tumorigenesis. As reported, tumor genomics of RNAi performed in a mouse model of hepatocellular carcinoma has identified DDX20 as a tumor suppressor [104]. MiRNAs can regulate the expression of target genes, and, while they reportedly function as suppressors or oncogenes in several tumors, their expression tends to be low in tumor tissue [105,114,115]. As mentioned above, miRNA-140-3p and miRNA-22, as miRNA, can inhibit NF-κ B activity, and DDX20 deletion in hepatocellular carcinoma (HCC) impaired this inhibitory effect of miRNA-140-3p and miRNA-22 [116]. Decreases in miRNA-140-3p and miRNA-22 expression affected the inhibitory effect of NCOA1 and NRIP1 on NF-κB activity. In addition, maintaining NF-κB activity led to inflammation, further exacerbating hepatocellular carcinoma [96]. In addition, miR-140-3p dysfunction promotes tumorigenesis by increasing the expression of its downstream target gene Dnmt1 and the methylation of CpG islands in the promoter region of metallothionein (MTs) and decreasing MT expression, consequently enhancing NF-κB signaling pathway activity [9]. Therefore, it is inferred that DDX20 can function as a tumor suppressor. Certain proteins, such as death-associated protein kinase 1 (DAPK), can inhibit the proteasomal degradation of DDX20, maintaining high DDX20 levels [98]. Elevated DDX20 levels inhibit hepatocellular carcinoma cell migration and invasion by regulating the NF-κB signaling pathway. Figure 5B demonstrates that DDX20 affects tumorigenesis and development through the NF-κB signaling pathway as a tumor suppressor.

Thus, DDX20 acts either as an oncogenic factor promoting tumor development or as a tumor suppressor inhibiting tumor progression. In conclusion, DDX20 plays a role in the regulation of hepatocellular carcinoma development through various miRNAs, including miRNA-140-3P and miRNA-22, as well as in breast cancer development through miRNA-222. DDX20 primarily functions by influencing the NF-κB signaling pathway. Beyond its roles in breast cancer, prostate cancer, and hepatocellular carcinoma, DDX20 has been implicated in several other tumors [117]. Table 2 outlines the expression and function of DDX20 across multiple tumors.

## 8. Functions in Viral Infection

Early research identified the capacity of DDX20 to interact with two vitally encoded nuclear antigens of the EB virus (EBV), EBNA2 and EBNA3C, and modulate the transcription of viral and cellular genes [68]. EBV is a lymphocryptovirus (LCV) herpesvirus that predominantly infects B lymphocytes and is noted for its ability to maintain long-term latent infections in the body while expressing a limited number of “latent” genes [119,120].

According to the report, EBNA2 can serve as a transcriptional activator of transformative viral and cellular genes by regulating two EBNA2-regulated viral promoters (TP1 and LMP/TP2 promoter) following its binding to the homologous promoter element RBPJkappa [121,122]. This promoter activation by EBNA2 is facilitated by cellular enhancer binding proteins and EBV nucleoproteins [123,124].

Later studies revealed that DDX20 can bind EBNA2 and survival motor neuron (SMN) proteins via its C-terminal structural domain, reporting that EBNA2 targets the spliceosome complex to release the SMN protein after binding DDX20, which ultimately acts as a coactivator in RNA polymerase II transcriptional complexes on the LMP1 promoter [125]. An additional finding suggests that, although RBPJkappa is necessary for EBNA2 transactivation, it is insufficient and it needs to be achieved through the EBNA2 and SMN proteins, underscoring the crucial role of DDX20 [125].

Conversely, EB nuclear antigen 3C (EBNA3C) is an important latent antigen that induces B lymphocyte immortalization in EBV through its contribution to viral pathogenicity. This is achieved via the putative bZIP structural domain at the N terminus of the protein and its interaction with cellular transcription factors RBPJkappa and HDAC1 to regulate transcriptional activation [126,127]. Additionally, EBNA3C facilitates the transcriptional reprogramming of various host cell genes, which is associated with the lengthy latency of EBV [128].

Similarly, DDX20 interacts with EBNA3C via its C-terminal structural domain. DDX20 can be stabilized by EBNA3C to form a complex with p53, which subsequently blocks p53-mediated transcriptional activity and apoptosis [67]. Moreover, based on related studies, EBNA2 and EBNA3C can reportedly disrupt the interaction between DDX20 and the transcription factor METS, thereby activating cellular proto-oncogenes [129]. This connection between DDX20 and EBV led to the initial discovery of a link between DDX20 and cancer.

In addition to EBV, DDX20 has significant connections with other viruses. DDX20 expression has previously been identified to vary in tumor tissues. In terms of changes in its expression upon viral infection, researchers reported that DDX20 is differentially expressed at distinct stages of human immunodeficiency virus type 1 (HIV-1) infection and found that DDX20 interacts with the HIV-1 coprotein Vpr as early as 2012 [76].

Further analysis of global miRNA and mRNA expression through microarray and quantitative reverse transcription polymerase chain reaction in well-characterized HIV-1 latently and actively infected cells revealed that DDX20 was significantly upregulated in the former [130]. Conversely, a downregulation of DDX20 expression was detected during HIV-1 replication. DDX20 was directly degraded by the vRNA translation product of HIV-1, the Vpr protein, via the DCAF1/DDB1/CUL4 E3 ubiquitin ligase-mediated degradation pathway [131]. Owing to these alterations in DDX20 expression, it is thought to play a part in HIV-1 replication. However, the precise mechanism underlying the potential inhibitory effect of DDX20 remains unclear, thus necessitating further research.

DDX20 can enhance Interferon regulatory factor 3 (IRF3) phosphorylation levels by promoting the interaction between TBK1 and IRF3, which further promotes IFN-β expression, ultimately inhibiting the replication of VSV and HSV-1 through INF-stimulated genes (ISG) [2]. This opens up avenues for investigating the role of DDX20 in innate immunity research. Thus, although the role of DDX20 in suppressing viral infection and innate immunity has not been extensively studied, it has great research potential.

## 9. Concluding Remarks

Since its identification as a member of the DEAD family of RNA-unwinding enzymes, the diverse functions of DDX20 have been gradually unveiled. Early discoveries indicated that DDX20 could modulate transcriptional processes by influencing transcription factors such as SF-1, PE-1/METS, and Egr2/Krox-20 [51,63,70]. However, no new findings have been reported in recent years regarding the impact of DDX20 on transcription and the underlying mechanisms of transcription factors. Conversely, extensive research has focused on the involvement of DDX20 in the SMN complex and snRNP assembly, where it contributes to post-transcriptional RNA splicing [44,45]. Notably, the localization of DDX20 in the cytoplasm undergoes changes within the SMN complex, owing to its function [44,45]. Moreover, the phosphorylation and sumoylation of DDX20 can affect its migration and functional performance [28,29]. Additionally, DDX20 may participate in RNA deconvolution and RNP recombination during miRNA maturation and target RNA recognition through its inherent deconvolution enzyme activity. These lines of evidence highlight the significant role of DDX20 in transcription and RNA processing.

Furthermore, early investigations revealed the involvement of DDX20 in the transcription of viral genes influenced by EBV nuclear antigens [68]. Recent studies have discovered the antiviral role of DDX20 in innate immunity against HSV infection, although the underlying mechanisms remain unclear. The field of viral infection and antiviral innate immunity offers substantial research potential. These functions of DDX20 are closely tied to its structural characteristics, with its C-terminal domain serving as a prerequisite for interaction with various proteins. The role of DDX20 in tumors has also received considerable attention as it functions not only as a double-sided tumor suppressor and promoter but also as a potential antitumor drug target and marker. Notably, the association of DDX20 with tumors mainly occurs through the NF-κB signaling pathway, and its diverse effects on the pathway reflect its differential impact on tumors [96,108]. Consequently, the study of DDX20 in the context of tumors is expected to remain a highly relevant and active field of research.

Given the close relationship between the functions of DDX20 and its structural attributes, a comprehensive understanding of the structure of DDX20 is crucial. Whether it interacts with transcription factors to regulate transcriptional processes or acts as a target protein in tumorigenesis, the C-terminal domain of DDX20 plays a vital role. Therefore, gaining better insights into the structure of DDX20 is imperative. Moreover, it is foreseeable that DDX20 can be developed as a potential drug target for antitumor treatments in oncological diseases. Furthermore, the specific mechanism through which DDX20 exerts its antiviral effect via innate immunity remains unclear, emphasizing the need for further research to unveil the mechanisms underlying virus replication. Such investigations can also establish a theoretical foundation for the development of relevant antiviral products.

## Figures and Tables

**Figure 1 molecules-28-07198-f001:**
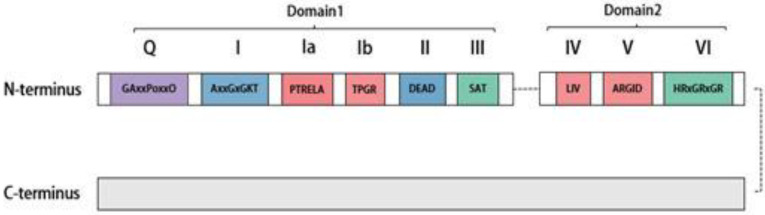
Structure of DEAD-box helicase DDX20. The upper part is the N-terminal section, depicted at the top, forms the helicase core region of Gemin3, comprised of two RecA-like helicase domains (domain 1 and domain 2). Structural Domain 1 encompasses six conserved motifs, while structural domain 2 includes three conserved motifs. The purple region denotes motif Q, which regulates ATPase activity; the blue region represents Motifs I and II, responsible for ATP binding and hydrolysis. The red region portrays Motifs Ia, Ib, IV, and V, associated with RNA substrate binding, while the green section illustrates Motifs III and VI, controlling communication between ATP-binding and RNA binding sites. The remaining portion of the figure represents the C-terminus of DDX20, which is involved in modulating RNA helicase activity, determining substrate specificity, and mediating interactions with other proteins.

**Figure 2 molecules-28-07198-f002:**
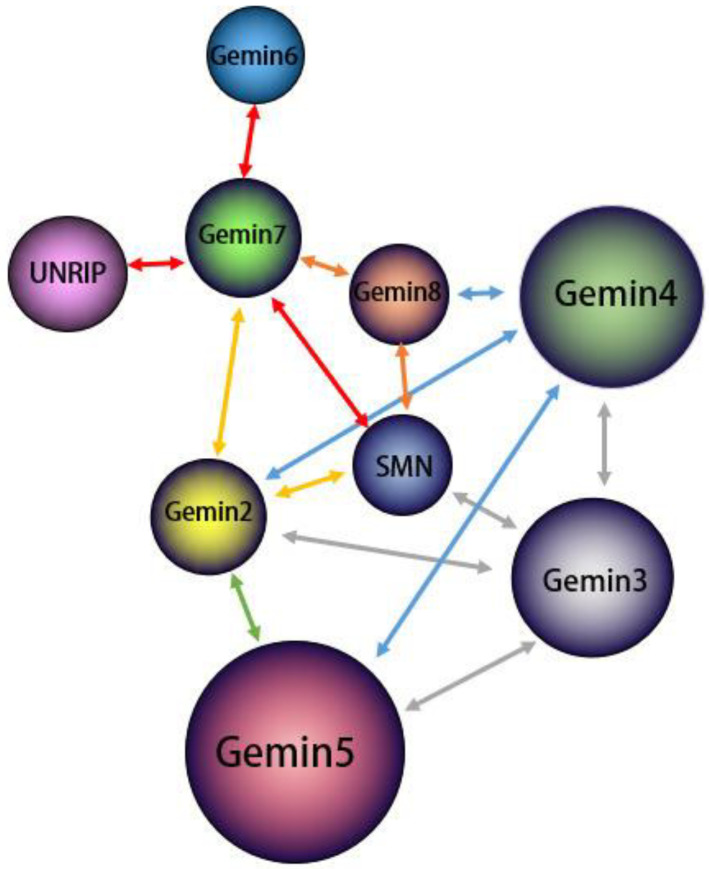
Structure and interactions within the SMN Complex. All arrows in the figure indicate the presence of interactions between proteins. The complete SMN complex comprises nine proteins, including Gemin3 as a core component. The stability of the SMN complex is dictated by interactions among its various components. As depicted by the gray arrows, Gemin3 engages with Gemin2, 4, and 5, as well as SMN proteins within the complex, forming the SMN complex’s core and enhancing its stability. Interactions among other components, demonstrated via various methods, are not elaborated here.

**Figure 3 molecules-28-07198-f003:**
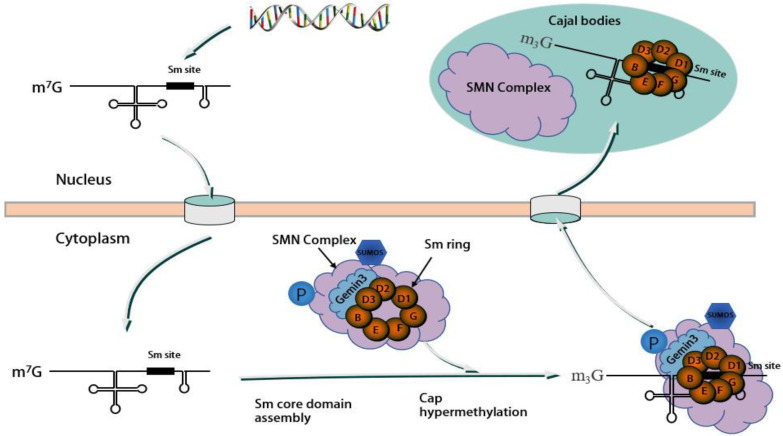
Assembly process of the spliceosomal snRNP. The snRNA is initially transcribed in the nucleus, undergoing processes such as splicing and capping before being transported into the cytoplasm with the aid of various factors (not depicted in the figure). Assisted by the SMN complex components (with Gemin3 interacting with SmB, SmD2, and SmD3), the Sm proteins form a heptameric ring and bind to the Sm site on the snRNA. Subsequently, the cytoplasmic snRNA is modified to develop an m3G cap structure before being re-transported to the nucleus as an snRNP, where it localizes in Cajal bodies. During this process, Gemin3 and other components undergo phosphorylation and sumoylation modifications to ensure the proper progression of the process.

**Figure 4 molecules-28-07198-f004:**
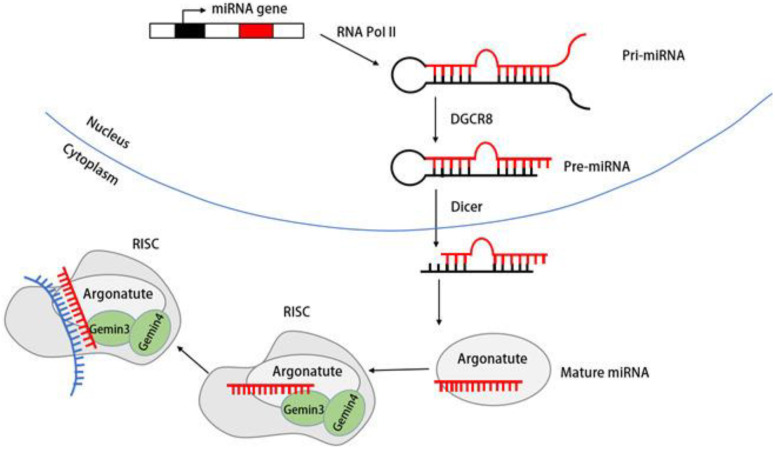
Gemin3 facilitates miRNA biogenesis. The process begins with the transcription of the miRNA gene, producing Pri-miRNA. The RNase III nucleic acid endonuclease then cleaves Pri-miRNA, generating pre-miRNA, which is subsequently transported to the cytoplasm. In the cytoplasm, mature miRNAs join forces with the Gemin3/Gemin4 complex, members of the Argonaute protein family, and other proteins, forming the RISC complex. Gemin3 plays a crucial role in supporting the biological functions of miRNAs.

**Figure 5 molecules-28-07198-f005:**
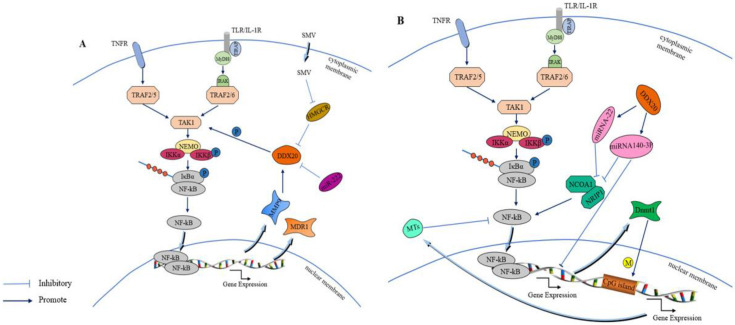
(**A**): DDX20 promotes the phosphorylation of TAK1 in the NF-κB signaling pathway, leading to the activation of the NF-κB signaling pathway. This activation increases the expression of MMP9, which in turn promotes the expression of DDX20, forming a complete DDX20-NF-κB-MMP9 axis. Additionally, simvastatin inhibits 3-hydroxy-3-methylglutaryl coenzyme A reductase, resulting in reduced DDX20 expression. (**B**): DDX20 plays a role in maintaining the normal functional performance of miRNA-140-3p and miRNA-22. Specifically, miRNA-140-3p downregulates the expression of Dnmt1, preventing Dnmt1-induced methylation of CpG islands in the promoter region of metallothionein MTs. This downregulation leads to increased expression of MTs and inhibition of the activation of the NF-κB signaling pathway. Additionally, miR-22 inhibits NF-κB activation by targeting the expression of NRIP1 and NCOA1. Furthermore, miRNA-140-3p also downregulates the expression of Dnmt1, preventing Dnmt1-induced methylation of CpG islands in the promoter region of metallothionein MTs, resulting in upregulation of MTs expression and inhibition of the activation of the NF-κB signaling pathway.

**Table 1 molecules-28-07198-t001:** Several factors interacting with the C-terminal of DDX20.

Interacting Protein	Key Function	Interaction Technique	Refs
SMN	RNP assembly function	GST-pulldown, Y2H, co-IP, genetic interaction	[23,42]
Gemin2	RNP assembly function	co-IP, genetic interaction	[42,66]
Gemin4	SMN complex member	Y2H, co-IP, genetic interaction	[42,43]
Gemin5	RNP assembly function	co-IP, genetic interaction	[42,66]
Sm B	snRNP component	GST-pulldown	[23]
Sm D2	snRNP component	GST-pulldown	[23]
Sm D3	snRNP component	Y2H, co-IP	[23]
pICln	RNP assembly function	Y2H, genetic interaction	[49]
Tgs1	snRNP cap	Y2H, genetic interaction	[49]
EBNA2	Epstein–Barr virus-encoded nuclear antigens	Y2H, co-IP	[67,68]
EBNA3C	Epstein–Barr virus-encoded nuclear antigens	Y2H, co-IP, GST-pulldown	[67,68]
p53	Tumor suppressor	GST-pulldown	[4,67]
TAK1	Regulator of NF-κB signaling pathway	co-IP	[69]
SF-1	Transcription factor	M2H, His-pulldown	[10,20]
METS/PE1	Transcription factor	Y2H, GST-pulldown	[63,64]
FOXL2	Transcription factor	Y2H, co-IP	[59,61]
Egr2 / Krox-20	Transcription factor	Y2H	[70]
AGO2/eIF2C2	RNA silencing	co-IP, GST-pulldown	[71,72,73]
HspB8/Hsp22	Heat shock protein	Y2H, co-IP	[74]
Olig2	Transcription factor	Y2H	[4]
FTZ-F1	Transcription factor	co-IP	[59]
MGF360-9L	Type I interferon inhibitors	LC-MS, co-IP	[75]
IRF3	Transcription factor	co-IP	[2]
Vpr	Accessory factors	AP-MS	[76]

**Table 2 molecules-28-07198-t002:** Expression and function of DDX20 in several tumors.

Cancer Type	DDX20 Expression	DDX20 Function	Mode of Action	Refs
Gastric cancer	Upward revision	In vitro promotion, high expression inhibition	Immune activation	[13]
Prostate cancer	Upward revision	Enhances the proliferation and metastasis of cancer cells	Activating the NF-κB signaling pathway	[11]
Oral squamous cell carcinoma	——	Predicts OSCC invasion forms	——	[118]
Colorectal cancer	Genetic variation	Presumptive markers of relapse	——	[6]
Hepatocellular carcinoma	Lower	Promotes hepatocellular carcinoma	Impaired miRNA140-3P function	[9,96]
Invasive breast cancer	Upward revision	Biomarker and metastasis-driving oncogene	DDX20–NF-κB–MMP9 axis Reduced miR-222	[69,111]
Triple-negative breast cancer	Upward revision	Biomarker and metastasis-driving oncogene	DDX20-NF-κB-MMP9 axis Reduced miR-222 TCF 4 and Wnt/beta-catenin signaling	[69,111,113]

## Data Availability

Not applicable.

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
