# Peer review of "DDX20: A Multifunctional Complex Protein"

_molecules, 2023, doi:10.3390/molecules28207198_

Round 1

Reviewer 1 Report

  1. The overall organization of the article needs to be improved for better flow and coherence. The authors should consider reordering sections to create a logical progression of the topic. For example, describe the structure and biogenesis of DDX20, its function in miRNA maturation and transcription factor suppression, followed by its role in tumorigenesis and viral replication.

  2. Schemes and diagrams should be added to explain the interaction between DDX20 and other proteins, as well as the pathways via which DDX20 acts on tumor development and viral replication. Proper citation should be included for referenced figures.

  3. In Line 65, specific miRNAs affecting NF-kB regulated by DDX20 should be summarized. 

The English language used in the article needs further refinement for clarity and readability.

Reviewer 2 Report

This article leads us to systematically recognize all the current studies on DDX20 protein, and the authors have highly summarized these studies into two parts, which enables us to easily understand the progress of research on DDX20. In addition, the pictures and tables drawn by the authors give us a better visualization of the research on DDX20 in a particular area. And, the authors also pointed out the research prospect of DDX20 in the article. It is an excellent article that systematically presents the research work about DDX20.

Howeverthere are some problems, which must be solved before it is considered for publication. If the following problems are well-addressed, this reviewer believes that the essential contribution of this paper are important for future research on DDX20. Specific issues are as follows:

1) "Tumor" and "cancer" are two completely distinct concepts. It is widely known that "tumors are classified into benign and malignant tumors, with only malignant tumors developing into cancer." Therefore, it is recommended to review the usage of "tumor" and "cancer" throughout the entire text, especially in the section titled "3. DDX20 Plays Different Roles in Tumors Through the NF-κB Signaling Pathway."

2) Throughout the article, numerous English abbreviations are introduced for the first time without their corresponding full forms. For instance, "REDOX and Wnt" in lines 134-136, and "TCF 4" in line 136. It is recommended that the full text be carefully proofread and that the full form of these abbreviations be provided.

3) In line 65.The abbreviation for "noncoding RNA" should be "nc RNA," while the full name for "miRNA" is "microRNA." Recommended revision.

4) In line 68.There are various expressions for "NF-κB signaling pathway" in the article, such as "NF-κB signaling" and "NF-κB pathway". It is recommended that the term "NF-κB signaling pathway" be used consistently throughout the article.

5) In line 73.The abbreviation "miRNA" for "microRNA" has been explained in the previous text; it is recommended to consistently use the abbreviation in the subsequent sections. Suggested revision.

6) In line 74.The expression " NF-κB pathway signaling" does not seem to be correct. It is suggested that " NF-κB pathway signaling" be replaced by " NF-κB signaling pathway".

7) In line 82. "tp53" and "p53" should not be used interchangeably, as "tp53" represents the gene name, whereas "p53" corresponds to the protein name.Suggested revision of this sentence.

8) In line 87. There are various expressions for "p53 signaling pathway" in the article, such as " p53 signaling" and " p53 pathway". It is recommended that the term " p53 signaling pathway" be used consistently throughout the article.

9) In lines 145-147.This sentence contradicts the preceding paragraph; it is suggested to rephrase this sentence. Emphasize listing specific tumor types mentioned in the referenced literature where the phenomenon of tumor suppression by miRNA is observed.

10) In line 164. Since the sentence mentions "DDX20 has been implicated in several other tumors.", it is suggested that a reference be inserted at the end of the sentence.

11) In line 187. The implication of "Conversely" is that "EBNA2" is not " an important latent antigen ", but this is not mentioned above. It is suggested that "Conversely" be replaced by "According to reports";

At the same time, it was suggested that the expressions " EB nuclear antigen 3C (EBNA3C) " and " EBNA 2 and EBNA 3C in line 171" should be amended.

12) Regarding "Figure 1-4.". There doesn't seem to be a reference to " Figure 1-4." in the article. Suggest proofreading;

Moreover, are the "images" in the article sourced from the "references"? If so, authors are requested to add "citation" after the "figure caption".

13) In lines 471-473. There are too many similar expressions in these two paragraphs. Additionally, the portion from "lines 506-515" doesn't fully serve the purpose of a "concluding paragraph" and appears somewhat lengthy. It is suggested that the author condense and integrate these two sections for conciseness.

Round 2

Reviewer 1 Report

I would recommend publishing the manuscript if the following can be addressed.

1. Some of the protein names in Figure 2 are too small to be see. Higher resolution image is required to clearly show the protein names.

The language is good in general and minor editing of English may be required.